# INSTRUCTZERO:
# EFFICIENT INSTRUCTION OPTIMIZATION FOR
# BLACK-BOX LARGE LANGUAGE MODELS

## ABSTRACT

Large language models (LLMs) are instruction followers but the performance varies under different instructions. It is challenging to create the best instruction, especially for black-box LLMs on which backpropagation is forbidden. Instead of directly optimizing the discrete instruction, we optimize a low-dimensional soft prompt applied to an open-source LLM to generate the instruction for the black-box LLM. In each optimization step of the proposed method INSTRUCTZERO, a soft prompt is converted into an instruction by the open-source LLM, which is then submitted to the black-box LLM for zero-shot evaluation, whose result is sent to Bayesian optimization to produce new soft prompts improving the zero-shot performance. We evaluate INSTRUCTZERO on different combinations of open-source LLMs and APIs including Vicuna and ChatGPT. INSTRUCTZERO outperforms SOTA auto-instruction methods across a variety of downstream tasks.

## 1   INTRODUCTION

Large Language Models (LLMs) (OpenAI, 2023a;b; Chowdhery et al., 2022) have recently gained widespread attention due to their remarkable capabilities in following instructions under both zero-shot and few-shot settings (Brown et al., 2020; Liu et al., 2023; Chen et al., 2023a). However, their performance is sensitive to the choice of instructions (Zhou et al., 2022; Honovich et al., 2022). For example, even paraphrasing a good instruction can lead to the failure of LLMs on certain tasks. It is still not clear when and how the instruction-following capability of LLMs can be generalized.

Instruction-following capability is essential to LLMs when used as an interface between humans and AI models, i.e., human users can instruct LLMs to solve complicated tasks by providing in-context instructions. "Prompt engineering" (Brown et al., 2020; Liu et al., 2023) usually relies on human experts' experience to craft instructions through a costly trial-and-error process. Hence, how to automate the instruction search or optimization for any given task is a critical open challenge. Unlike soft prompts, instruction is composed of discrete words or sentences that are difficult to optimize in a continuous space. To create a human-interpretable and task-relevant instruction, we have to address combinatorial optimization with complex structural constraints. Moreover, the most powerful instruction-following LLMs, e.g., ChatGPT (OpenAI, 2023a) and GPT-4 (OpenAI, 2023b), are black boxes. Given their APIs only, it is infeasible to develop gradient-based optimization that requires back-propagation through these models.

In this paper, we propose an effective and efficient approach "INSTRUCTZERO" to tackle the zeroth-order combinatorial optimization of instructions to API LLMs (Chen et al., 2017; Wang et al., 2018; Schrijver et al., 2003; Wolsey & Nemhauser, 1999). Instead of directly optimizing the instruction, INSTRUCTZERO optimizes a soft prompt appended to a few exemplars of the target task, steering an open-source LLM (e.g., LLaMA (Touvron et al., 2023), Stanford Alpaca, Vicuna), to generate a human-readable and task-relevant instruction in an in-context learning manner. The instruction is then submitted to the black-box LLM for zero-shot evaluation on the target task, whose performance is used to guide the optimization of the soft prompt toward generating better instructions.

We formulate the soft prompt optimization as a form of latent space Bayesian Optimization (BO), which aims to maximize the zero-shot performance as a black box function. It estimates the black-box objective using each explored soft prompt and its zero-shot performance as an input-output sample, with a kernel relating all samples. The mean and variance of the estimation controls the exploration-exploitation of the soft prompts. To align the soft prompt optimization with the search in instruction space, we develop an instruction-coupled kernel to align the two spaces' kernels. Thereby, optimizing

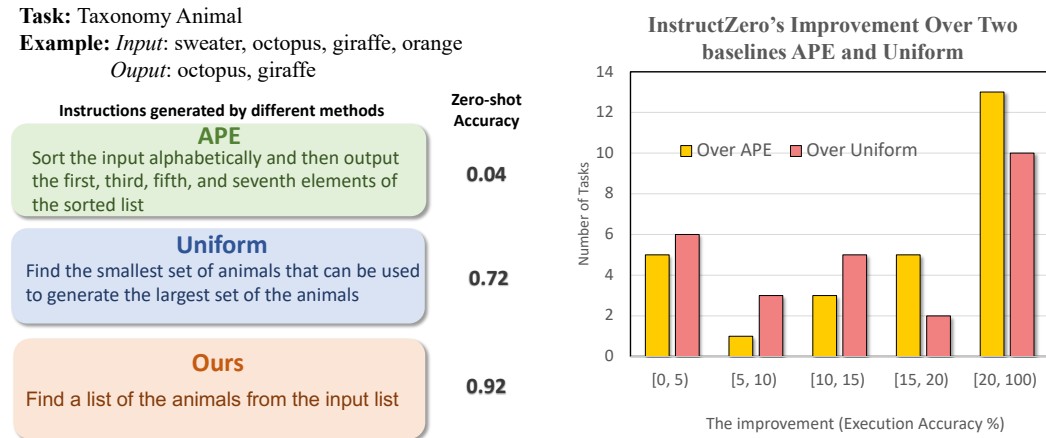

Figure 1: **Comparison** between INSTRUCTZERO and two baselines, i.e., APE (Zhou et al., 2022) and uniform sampling (defined in baselines of Section 4.1). **Left:** INSTRUCTZERO generate a more precise instruction leading to better performance (higher execution accuracy). **Right:** Histogram of INSTRUCTZERO's improvement over APE and Uniform on 32 tasks. INSTRUCTZERO achieves a significant improvement between $[20\%, 100\%]$ in terms of accuracy on a majority of evaluated tasks. The task is to pick out the animals from the list.

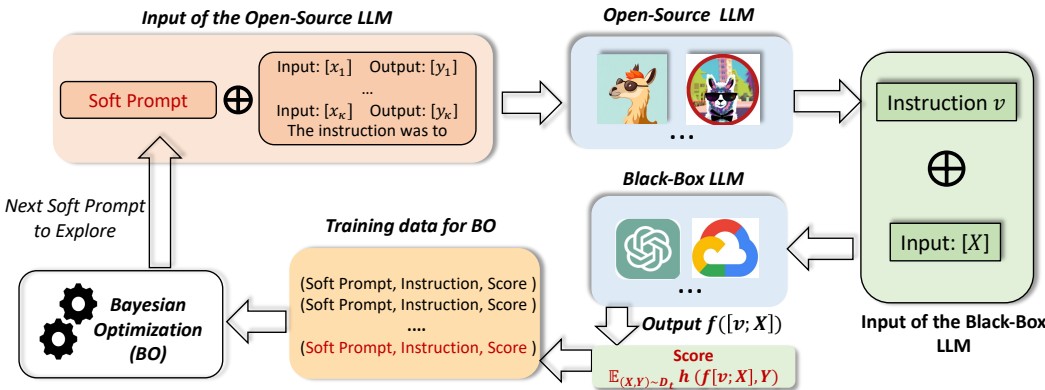

Figure 2: Pipeline of INSTRUCTZERO. On each iteration, a soft prompt and a few exemplars of the target task are sent to the open-source LLM for generating an instruction, which then prompts the black-box LLM to produce answers to target-task queries. The score (e.g., accuracy) of the answers and the soft prompt is added as new training data for BO, which updates its posterior about the objective (score) and produces a new soft prompt to explore in the next iteration. Both LLMs are frozen.

the low-dimensional soft prompt leads to an efficient search for optimal instruction in the sparse and highly structured textual space.

We evaluate INSTRUCTZERO on a combination of SOTA open-source LLM and black-box LLM, i.e., 13-B Vicuna and GPT-3.5-turbo (ChatGPT). Experimental results show that ChatGPT's performance is significantly improved when using the instructions optimized by INSTRUCTZERO: It achieves SOTA results on 32/32 tasks from BIG-Bench. As a case study, we visualize an instruction optimization process of INSTRUCTZERO and the instructions generated in every step. INSTRUCTZERO, even using much weaker Vicuna models, outperforms non-optimization methods Zhou et al. (2022) that use ChatGPT generating instructions.

## 2 INSTRUCTION OPTIMIZATION

### 2.1 PROBLEM FORMULATION

We study how to optimize an instruction $v$ applied to a black-box LLM $f(\cdot)$ to address a task with input query $X$. In particular, the optimization objective aims to maximize the output $f([v; X])$'s performance $h(f([v; X]), Y)$, which uses a score produced by an evaluation metric $h(\cdot, \cdot)$ comparing $f([v; X])$ and the ground truth $Y$. Hence, the optimization of instruction $v \in \mathcal{V}$ can be formulated as maximizing the expected score $h(f([v; X]), Y)$ for an example $(X, Y)$ drawn from the data

distribution $\mathcal{D}_t$ of task-$t$, i.e.,

$$\max_{v \in \mathcal{V}} \ \mathbb{E}_{(X,Y) \sim \mathcal{D}_t} h(f([v; X]), Y). \tag{1}$$

Unfortunately, Eq. (1) is notoriously challenging or practically infeasible because it is **(1) Combinatorial optimization** with complicated structural constraints: the instruction $v$ that can be taken by black-box LLMs such as ChatGPT and GPT-4 is a combination of discrete tokens that have to comprise human-readable and task-relevant sentence(s). Thus, its optimization space $\mathcal{V}$ is high-dimensional, discrete, and highly structured due to semantic constraints. In general, there do not exist efficient optimization algorithms in such a space; and **(2) Black-box optimization**: the black-box LLM $f(\cdot)$ makes the objective as a black-box function. Users are only allowed to input texts to $f(\cdot)$ and only obtain textual outputs. Hence, backpropagation through $f(\cdot)$ and any gradient-based algorithm to optimize the objective cannot be applied.

Instead of optimizing the instruction $v$ in the original space $\mathcal{V}$, the key idea of INSTRUCTZERO is to optimize a soft prompt $\boldsymbol{p}$ applied to an open-source LLM $g(\cdot)$, which converts $\boldsymbol{p}$ to a human-readable and task-relevant instruction $v$ via in-context learning with $\kappa$ exemplars $(x_i, y_i)_{i=1}^{\kappa}$ drawn from the target task. The instruction $v$ is then applied to the black-box LLM $f(\cdot)$ to produce zero-shot prediction $f([v; X])$. The zero-shot performance score $h(f([v; X]), Y)$ on target task data $(X, Y) \sim \mathcal{D}_t$ is collected to estimate the objective function in Eq. (1) by Bayesian optimization (BO), which proposes new soft prompts for generating better instructions.

The pipeline of INSTRUCTZERO is illustrated in Fig. 2, where the open-source LLM can be LLaMA, Alpaca, Vicuna, etc., and the black-box LLM can be ChatGPT (OpenAI, 2023a), GPT-4 (OpenAI, 2023b), Claude, PaLM-2 (Google, 2023), etc. By generating the instruction using an open-source LLM, INSTRUCTZERO reduces the challenging instruction optimization to a feasible black-box optimization of a soft prompt in a low-dimensional space, which can be addressed by latent space Bayesian optimization. The complete procedure is provided in Algorithm 1.

## 2.2 FROM STRUCTURED COMBINATORIAL SEARCH TO LOW-DIMENSIONAL CONTINUOUS OPTIMIZATION

INSTRUCTZERO, as shown in Fig. 2, applies an open-source LLM $g(\cdot)$ to generate instructions $v$ via in-context learning. Specifically, we concatenate a soft-prompt $\boldsymbol{p} \in \mathbb{R}^{d'}$ (a $d'$-dimensional vector) with $\kappa$ input-output exemplars $(x_i, y_i)_{i=1}^{\kappa}$ (represented by their token embeddings) drawn from the task's distribution $\mathcal{D}_t$ as input to the open-source LLM to generate an instruction $v = g([\boldsymbol{p}; x_{1:\kappa}])$ for the black-box LLM $f(\cdot)$. Therefore, the combinatorial instruction optimization in Eq. (1) can be reframed as a more feasible continuous optimization below.

$$\max_{\boldsymbol{p} \in \mathbb{R}^{d'}} \mathbb{E}_{(X,Y) \sim \mathcal{D}_t} h(f([v; X]), Y), \ \ s.t. \ v = g([\boldsymbol{p}; (x_i, y_i)_{i=1}^{\kappa}]), \tag{2}$$

**Dimension Reduction.** Though we reduce the original instruction optimization to continuous optimization of a soft prompt $\boldsymbol{p}$, it still needs to solve a black-box optimization due to the black-box LLM $f(\cdot)$ in the objective of Eq. (2). Unfortunately, as input tokens to an open-source LLM, $\boldsymbol{p}$ usually has dimensionality too high (e.g., thousands for Vicuna) to be handled by existing black-box optimization approaches. Hence, we instead optimize a lower-dimensional vector $\boldsymbol{p} \in \mathbb{R}^d$ where $d \ll d'$ and project it to $\mathbb{R}^{d'}$ using a simple random projection $A\boldsymbol{p}$ as input tokens to $g(\cdot)$, where each entry of the matrix $A \in \mathbb{R}^{d \times d'}$ is sampled from Normal or Uniform distribution (Wang et al., 2016). This is based on: (1) the random projection is distance-preserving according to Johnson-Lindenstrauss Lemma (Kleinberg, 1997), which leads to comparable kernel similarities before and after the random projection, i.e., $k(\boldsymbol{p}_i, \boldsymbol{p}_j) \approx k(A\boldsymbol{p}_i, A\boldsymbol{p}_j)$, so BO in the original space and dimension-reduced space are consistent; (2) Thanks to in-context learning capability of the open-source LLM, when concatenated with $\kappa$ exemplars, low-dimensional soft prompt suffice to produce rich, diverse, and task-relevant instructions as candidates. Therefore, by replacing $\boldsymbol{p}$ in Eq. (2) with $A\boldsymbol{p}$, the instruction optimization in Eq. (1) is reduced to maximization of a black-box function $H(\boldsymbol{p})$ in a low-dimensional space $\mathbb{R}^d$, i.e.,

$$H(\boldsymbol{p}) \triangleq \mathbb{E}_{(X,Y) \sim \mathcal{D}_t} h(f([v; X]), Y), \ \ v = g([A\boldsymbol{p}; (x_i, y_i)_{i=1}^{\kappa}]). \tag{3}$$

## 3 BAYESIAN OPTIMIZATION WITH INSTRUCTION-COUPLED KERNEL

In the previous section, we reduced the instruction generation problem to a black-box optimization in a low-dimensional space, i.e., $\max_{\boldsymbol{p} \in \mathbb{R}^d} H(\boldsymbol{p})$, which can be addressed by Bayesian optimization

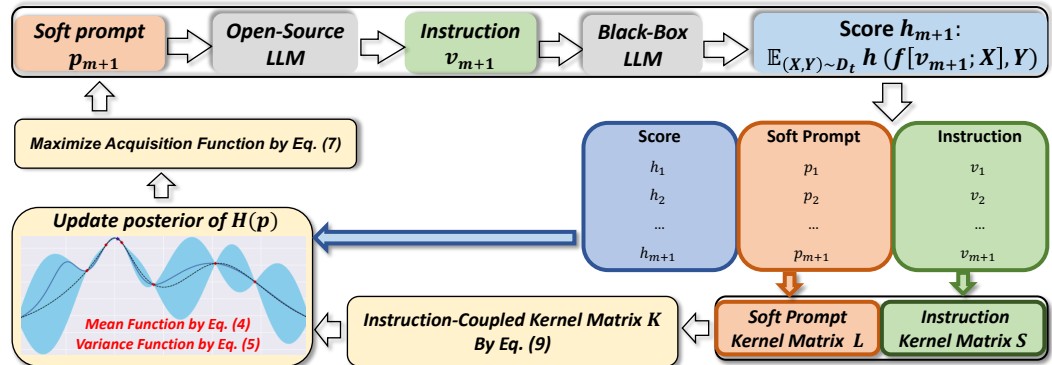

Figure 3: The pipeline of Bayesian optimization in INSTRUCTZERO proposed in Section 3.

(BO). Specifically, BO aims to estimate the black-box objective $H(\boldsymbol{p})$ and finds its maximum; it keeps updating a posterior of $H(\cdot)$ based on collected $(\boldsymbol{p}, H(\boldsymbol{p}))$ pairs and exploring new soft prompts $\boldsymbol{p}$ until the largest $H(\boldsymbol{p})$ converges to a maximum. To evaluate $H(\boldsymbol{p})$ on a soft prompt $\boldsymbol{p}$ and its generated instruction, we average the zero-shot performance $h(f([v; X]), Y)$ on a validation set.

## 3.1 BAYESIAN OPTIMIZATION OF SOFT PROMPT

We apply the commonly used Gaussian Process (GP) as the prior for the black-box objective $H(\cdot)$. A GP prior can be specified by a mean function $\mu(\cdot) = 0$ and a covariance function (i.e., kernel function) $k(\cdot, \cdot)$. Given $m$ soft prompts $\boldsymbol{p}_{1:m} \triangleq \{\boldsymbol{p}_1, \cdots, \boldsymbol{p}_m\}$ and their evaluation $H_{1:m} \triangleq [H(\boldsymbol{p}_1), a \cdots, H(\boldsymbol{p}_m)]$ collected in all previous BO steps, the estimated posterior of $H(\cdot)$ is updated as a Gaussian $\mathcal{N}(\mu(\cdot), \sigma^2(\cdot))$ with mean function $\mu(\cdot)$ and variance function $\sigma^2(\cdot)$ defined as, $\forall \boldsymbol{p} \in \mathbb{R}^d$,

$$\mu(\boldsymbol{p}) \triangleq \boldsymbol{k}(\boldsymbol{K} + \eta^2 \boldsymbol{I})^{-1} H_{1:m}, \tag{4}$$

$$\sigma^2(\boldsymbol{p}) \triangleq k(\boldsymbol{p}, \boldsymbol{p}) - \boldsymbol{k}^\top (\boldsymbol{K} + \eta^2 \boldsymbol{I})^{-1} \boldsymbol{k}, \tag{5}$$

where $\boldsymbol{k} = [k(\boldsymbol{p}, \boldsymbol{p}_1), \cdots, k(\boldsymbol{p}, \boldsymbol{p}_m)]$ and constant $\eta$ measures the noise levels of observations.

Expected improvement acquisition function (EI) measures the improvement of a candidate soft prompt over the best soft prompt in terms of the objective value, i.e., $\max\{0, H(\boldsymbol{p}) - \max_{i \in [m]} H(\boldsymbol{p}_i)\}$, and takes the improvement's expectation w.r.t. $H(\boldsymbol{p})$, which is a random variable with a distribution defined by the posterior of $H(\cdot)$. Therefore, EI $u(\cdot)$ is defined as, $\forall \boldsymbol{p} \in \mathbb{R}^d$,

$$u(\boldsymbol{p}) = \mathbb{E}_{H(\boldsymbol{p}) \sim \mathcal{N}(\mu(\boldsymbol{p}), \sigma^2(\boldsymbol{p}))} \left[ \max \left\{ 0, H(\boldsymbol{p}) - \max_{i \in [m]} H(\boldsymbol{p}_i) \right\} \right], \tag{6}$$

and BO explores the next soft prompt $\boldsymbol{p}_{m+1}$ maximizing the acquisition function:

$$\boldsymbol{p}_{m+1} \in \arg\max_{\boldsymbol{p} \in \mathbb{R}^d} u(\boldsymbol{p}). \tag{7}$$

The new soft prompt $\boldsymbol{p}_{m+1}$ is converted to an instruction $v_{m+1}$ by the open-source LLM $g(\cdot)$, i.e., $v_{m+1} = g([A\boldsymbol{p}_{m+1}; (x_i, y_i)_{i=1}^\kappa])$, and $v_{m+1}$ is applied to the black-box LLM for evaluating its zero-shot performance on the target task, i.e., $H(\boldsymbol{p}_{m+1})$. BO then augments its collected training data $(\boldsymbol{p}_{1:m}, H_{1:m})$ with $(\boldsymbol{p}_{m+1}, H(\boldsymbol{p}_{m+1}))$ and the procedure in Eq. (4)-(7) is repeated until convergence. The BO pipeline in INSTRUCTZERO is illustrated in Fig. 3.

## 3.2 INSTRUCTION-COUPLED KERNEL

The choice of kernel $k(\cdot, \cdot)$ in BO is critical to the performance of black-box optimization since it defines both the mean and variance of the posterior and thus guides the whole optimization process. In INSTRUCTZERO, although we conduct BO in the latent space of soft prompts, the goal is to optimize instructions in the instruction space $\mathcal{V}$. Hence, the kernel applied in the latent space should reflect the similarity of the generated instructions in the target task. In other words, we need to align the latent space kernel with the instruction similarity. To this end, we develop a novel instruction-coupled kernel inspired by (Deshwal & Doppa, 2021a).

Without loss of generality, we assume that BO in all previous steps has already explored $m$ soft prompts $\boldsymbol{p}_{1:m}$, which were converted to $m$ instructions $\boldsymbol{v}_{1:m} = \{v_1, v_2, ..., v_m\}$ via the open-source LLM. To measure the correlation between two soft prompts in the latent space $\mathbb{R}^d$, we choose a kernel

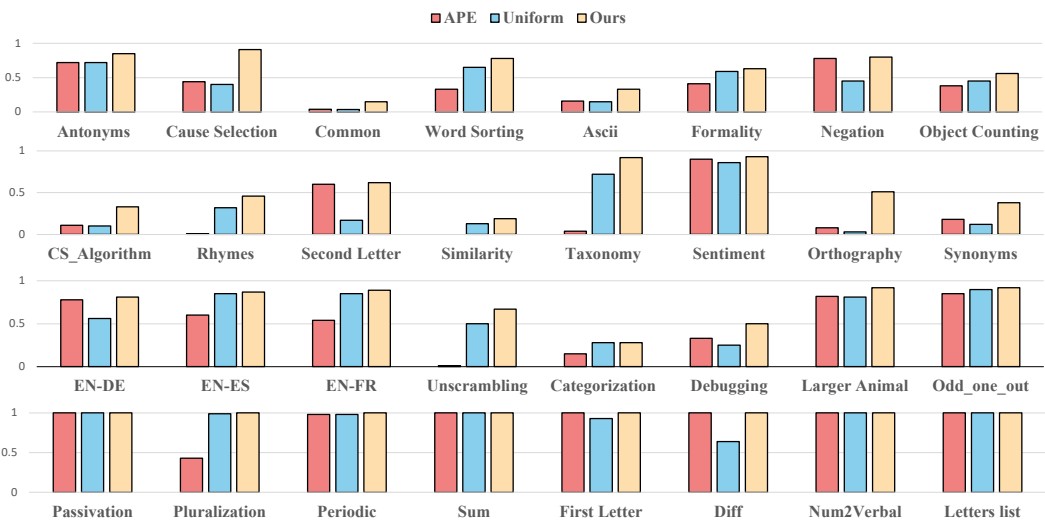

Figure 4: Zero-shot test accuracy on 32 tasks from (Honovich et al., 2022). INSTRUCTZERO achieves the best performance on all 32 out of 32 tasks among the three evaluated approaches.

function $l(\cdot, \cdot) : \mathbb{R}^d \times \mathbb{R}^d \to \mathbb{R}$, whose common options include Matern or Squared Exponential kernels. Applying $l(\cdot, \cdot)$ to $\boldsymbol{p}_{1:m}$ produces a kernel matrix $\boldsymbol{L} \in \mathbb{R}^{m \times m}$. To measure the similarity between two instructions in the target task, we define another kernel function $s(\cdot, \cdot) : \mathcal{V} \times \mathcal{V} \to \mathbb{R}$, for example, the similarity between their zero-shot predictions on target task data, i.e.,

$$s(v_i, v_j) = \mathbb{E}_{X \sim \mathcal{D}_t} \left[ \text{sim}(f([v_i; X]), f([v_j; X])) \right], \tag{8}$$

where $\text{sim}(\cdot, \cdot)$ is a similarity of the predictions for the tasks, e.g., exact match, F1, or BLEU score. Applying $s(\cdot, \cdot)$ to $\boldsymbol{v}_{1:m}$ produces a kernel matrix $\boldsymbol{S} \in \mathbb{R}^{m \times m}$. We propose an instruction-coupled kernel function by combining the two kernels $l(\cdot, \cdot)$ and $s(\cdot, \cdot)$ in the following manner.

$$\boldsymbol{K}_{i,j} = k(\boldsymbol{p}_i, \boldsymbol{p}_j) = \boldsymbol{l}_i^\top \boldsymbol{L}^{-1} \boldsymbol{S} \boldsymbol{L}^{-1} \boldsymbol{l}_j \tag{9}$$

where $\boldsymbol{l}_i \triangleq [l(\boldsymbol{p}_i, \boldsymbol{p}_1), \cdots, l(\boldsymbol{p}_i, \boldsymbol{p}_m)]$ and $\boldsymbol{l}_j \triangleq [l(\boldsymbol{p}_j, \boldsymbol{p}_1), \cdots, l(\boldsymbol{p}_j, \boldsymbol{p}_m)]$. The proposed kernel preserves the instruction similarity in the soft prompt space: when applied to soft prompts $\boldsymbol{p}_{1:m}$, the resulted kernel matrix $\boldsymbol{K}$ exactly recovers the instruction matrix $\boldsymbol{S}$ because $\boldsymbol{K} = \boldsymbol{L}\boldsymbol{L}^{-1}\boldsymbol{S}\boldsymbol{L}^{-1}\boldsymbol{L} = \boldsymbol{S}$ according to Eq. (9). For new soft prompts $\boldsymbol{p} \notin \boldsymbol{p}_{1:m}$, the instruction-coupled kernel in Eq. (9) operates as a smooth extrapolation kernel. Therefore, by combining the two spaces' kernels, the proposed kernel aligns BO in the latent space $\mathbb{R}^d$ of soft prompts (Eq. (3)) with the instruction optimization (Eq. (1)) in the combinatorial and structured space $\mathcal{V}$. Fig. 3 shows when the kernel matrices are computed in the BO pipeline of INSTRUCTZERO.

---

**Algorithm 1:** INSTRUCTZERO

**input** : Exemplars $(x_i, y_i)_{i=1}^\kappa$ and a validation set $D_t$ of target task-$t$; open-source LLM $g(\cdot)$, black-box LLM $f(\cdot)$, maximal steps $T$; random matrix $A \in \mathbb{R}^{d \times d'}$

**initialize :** $\boldsymbol{p}_1 \sim \text{uniform}(-\tau, \tau)^d$ in $\mathbb{R}^d$; $m \leftarrow 1$, $\boldsymbol{p}_{1:0} \leftarrow \emptyset$, $v_{1:0} \leftarrow \emptyset$, $h_{1:0} \leftarrow \emptyset$

1 **while** *not converge and $m \le T$* **do**

2      Compute input prompt $A\boldsymbol{p}_m$ from low-dimensional soft prompt $\boldsymbol{p}_m$;

3      Generate instruction $v_m = g([A\boldsymbol{p}_m; (x_i, y_i)_{i=1}^\kappa])$ by the open-source LLM $g(\cdot)$;

4      Evaluate zero-shot score $h_m = \sum_{(X,Y) \in D_t} h(f([v_m; X]), Y)$ on the black-box LLM $f(\cdot)$;

5      Save data: $\boldsymbol{p}_{1:m} \leftarrow \boldsymbol{p}_{1:m-1} \cup \{\boldsymbol{p}_m\}$, $v_{1:m} \leftarrow v_{1:m-1} \cup \{v_m\}$, $h_{1:m} \leftarrow h_{1:m-1} \cup \{h_m\}$;

6      Update the instruction-coupled kernel function $k(\cdot, \cdot)$ and matrix $\boldsymbol{K}$ for $\boldsymbol{p}_{1:m}$ by Eq. (9);

7      Update the mean and variance function of BO in Eq. (4)-(5) using $k(\cdot, \cdot)$ and $\boldsymbol{K}$;

8      Find the next prompt $\boldsymbol{p}_{m+1}$ maximizing the acquisition function $u(\boldsymbol{p})$ in Eq. (6);

9      $m \leftarrow m + 1$;

10 **end**

**output** : The best instruction $v_{i^*}$ so far with $i^* \in \arg\max_{i \in [m]} h_i$

---

## 4 EXPERIMENTS

In this section, we evaluate INSTRUCTZERO as a tool to find an instruction that steers a black-box LLM towards a desired downstream behavior on a target task. Extensive experiments demonstrate that our method could effectively generate instructions that enhance task performance while achieving predictions on par with or even superior to those created by previous methods. Moreover, IN-STRUCTZERO produces instructions that sometimes reveal valuable tricks for optimal prompting that could be subsequently applied to new tasks.

### 4.1 TASKS, DATASETS, BASELINES, AND IMPLEMENTATION

**Tasks.** We assess the effectiveness of zero-shot in-context learning on instruction tasks proposed in (Honovich et al., 2022), including all 24 tasks used in previous auto-instruction work (Zhou et al., 2022). We further add 8 extra tasks to enrich the benchmark for evaluating all methods in more comprehensive scenarios spanning many facets of language understanding. We provide detailed descriptions of each task in the Appendix. Training-set examples can be used for instruction optimization but the final instruction $p^*$ is evaluated on a held-out test set. Zero-shot performance $H(p)$ on the test set is reported.

**Baselines.** We compare INSTRUCTZERO with two baseline methods: (1) **APE** (Zhou et al., 2022), which generates instructions using a more powerful LLM (i.e, ChatGPT[1]) than the open-source LLM in INSTRUCTZERO; and (2) **Uniform** (pure exploration), which uses the same models as INSTRUCTZERO and draws the same total number of soft prompts by uniform sampling without iterative BO procedure.

**Score Function.** In the experiments, we use a simple 0-1 loss as the score function $h(\cdot, \cdot)$, i.e, $h(f([v; X]), Y) = 1$ if $f([v; X]) = Y$, otherwise $h(f([v; X]), Y) = 0$. So the score $h_{1:m}$ in Algorithm 1 computes execution accuracy by averaging $h(f([v; X]), Y)$ over all validation examples $(X, Y) \in D_t$. A more fine-grained score can be the log-likelihood of the ground-truth answer under instruction $v$ and input $X$. It is worth noting that the choice of score function depends on the outputs provided by the black-box LLM, e.g., GPT3 returns the log probabilities of the most likely tokens [2] while ChatGPT only offers access to the generated answer [3]. Since we use ChatGPT as the black-box LLM, $h_{1:m}$ represents execution accuracy in our experiments.

**Implementation Details.** We implement IN-STRUCTZERO as illustrated in Fig. 2 with Vicuna and ChatGPT as the open-source LLM and API LLM, respectively. For each task, we draw $\tau = 5$ and 20 samples from the training set as the exemplars and validation set $D_t$, respectively. For the number of tokens in soft prompts, we search for the best value among $\{3, 5, 10\}$ based on the validation set performance. We draw entries of the random projection matrix $A$ from a uniform distribution between $[-1, 1]$. The dimensionality $d$ of $p$ is set to 10. In experiments, we apply a mini-batch version of INSTRUCTZERO that explores 25 soft prompts in every iteration. The only major change required

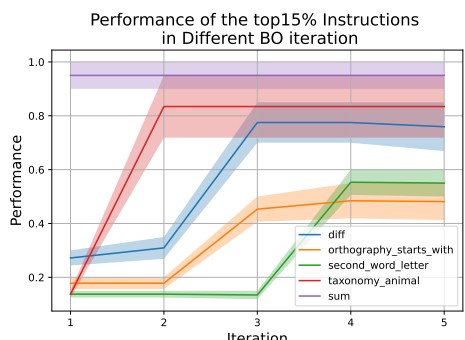

Figure 5: Top-15% instructions after every iteration (1-5) of INSTRUCTZERO on five tasks.

is to select the top-25 soft prompts with the largest $u(p)$ instead of maximizing Eq. (7) in Line 8 of Algorithm 1. We utilized an evolutionary search algorithm CMA-ES (Hansen, 2016) as the optimizer to find the top soft prompts. All the training and tests are conducted on a single NVIDIA RTX A6000 GPU card.

### 4.2 MAIN RESULTS

Fig. 4 reports the zero-shot test accuracy of ChatGPT when using instructions generated by APE, Uniform, and INSTRUCTZERO for 32 tasks. On easy tasks such as "Letters List" and "Sum", INSTRUCTZERO is comparable to APE which has already achieved perfect execution accuracy (i.e., 1.0). On the other hand, INSTRUCTZERO exhibits superior performance on challenging tasks such as

---

[1]GPT-3 was used in the original APE model but we re-evaluated it using the more powerful ChatGPT.

[2]https://platform.openai.com/docs/api-reference/completions/create

[3]https://platform.openai.com/docs/api-reference/chat/create

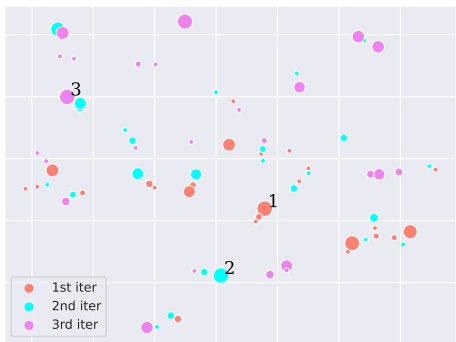

**Task:** Stronger animal
**Example:** *Input*: whale shark, dog
*Ouput*: whale shark

| | Instruction Generated by InstructZero | Accuracy |
|---|---|---|
| 1 | The instruction was to find the most dangerous animal in the zoo. | 0.65 |
| 2 | The instruction was to find out which animal is stronger between two animals. | 0.8 |
| 3 | The instruction was to input a animal and a animal into the system, and the system would output the stronger animal. | 1.0 |

Figure 6: The task is to write the stronger animals. **Left:** Soft prompts selected by INSTRUCTZERO in three consecutive iterations (2D embedding by t-SNE). Colors denote different iterations and a larger circle refers to a higher objective value (zero-shot validation accuracy). Numbers highlight the best soft prompt per iteration. **Right:** instructions generated by the best soft prompt per iteration and the associated validation accuracy.

"Unscrambling" and "Taxonomy Animal" where APE struggles. Fig. 1 (right) reports the histograms for the improvement of INSTRUCTZERO over the two baselines on all tasks except those easy ones on which both baseline and INSTRUCTZERO achieve (100%) test accuracy. Overall, the results demonstrate that instructions generated by INSTRUCTZERO significantly outperform those produced by the other two baselines by a large margin. We also summarize the best instruction created by INSTRUCTZERO for each task in the Appendix[4].

Fig. 5 shows the zero-shot accuracy of the top-15% instructions after each iteration of INSTRUCTZERO. On most tasks, the accuracy consistently improves over iterations, indicating an effective optimization process. Nonetheless, on easy tasks such as "Sum", the best instruction was identified in the very first iteration and thus further optimization was unnecessary.

### 4.3 ABLATION STUDY

| Task | Manual | w/o Manual | INSTRUCTZERO |
|---|---|---|---|
| Cause_and_effect | 0.36 | 0.56 | **0.91** |
| Negation | 0.27 | 0.01 | **0.80** |
| Translation_en-fr | 0.02 | 0.47 | **0.89** |
| Sum | 0.00 | 0.00 | **1.00** |
| Formality | 0.59 | 0.31 | **0.63** |
| Letters_list | 0.00 | 0.15 | **1.00** |
| Larger_Animal | 0.49 | 0.81 | **0.91** |

Table 1: **Ablation study.** Execution accuracy (higher is better) of the instructions obtained by INSTRUCTZERO and two baselines: (1) Manual: input to open-source LLM is exemplars $(x_i, y_i)_i^\kappa$ with the manual prompt; (2) w/o Manual: input to open-source LLM is exemplars $(x_i, y_i)_i^\kappa$ only.

To verify the effectiveness of optimization in INSTRUCTZERO, we compare it against two alternatives: (1) **Manual.** As illustrated in Fig. 7 shows, we replace the INSTRUCTZERO-optimized $p^*$ with a meta-prompt handcrafted by humans (used in APE (Zhou et al., 2022)) for instruction generation but keeps all the other parts the same in the test-setting for INSTRUCTZERO; and (2) **w/o Manual.** we further remove any prompt and solely use the $\kappa$ exemplars as input to generate instruction $v$. The comparison results are reported in Tab. 1, which shows a large improvement when using the soft prompt optimized by INSTRUCTZERO when compared to the two baselines. For example, on task "Letters List", INSTRUCTZERO achieves 100% accuracy while Manual Prompt is 0%. The improvement indicates that the optimized soft prompt plays a substantial role in instruction generation for better zero-shot performance on downstream tasks and BO in INSTRUCTZERO is effective in finding the optimal soft prompt.

---

[4]We report more results in Appendix: (1) INSTRUCTZERO's performance on other combinations of open-source LLM + API LLM; (2) INSTRUCTZERO's comparison to human written instruction. APE Zhou et al. (2022) shows advantages of their instructions over humans' and ours are better than APE.

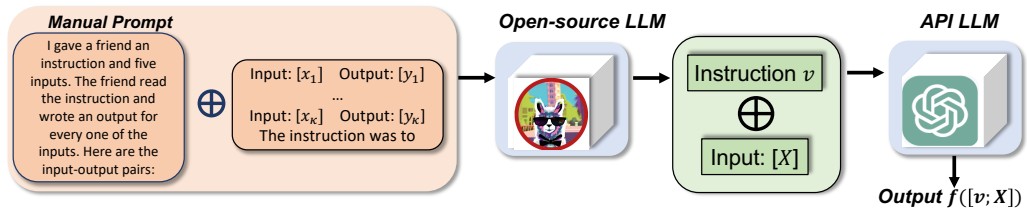

Figure 7: **Ablation study baseline.** Manual prompt in APE (Zhou et al., 2022) replaces the INSTRUCTZERO-optimized soft prompt used to generate instructions.

### 4.4 CASE STUDY

Fig. 6 visualizes the soft prompts explored by IN-STRUCTZERO over three BO iterations. It shows how the score of the best soft prompt improves over time and the efficient exploration-exploitation conducted by the latent space BO. The instructions generated using the best soft prompt in each iteration are given in the right of Fig. (6), which shows a progressive improvement of the instruction quality in terms of clarity, details, and task relevance. In Fig. 1 and 8, we compare the instructions generated by the three methods, i.e., Uniform, APE, and INSTRUCTZERO, for the same set of tasks. While both APE and Uniform can produce reasonable instructions, they exhibit notable drift from the task description. For instance, in Fig. 1,

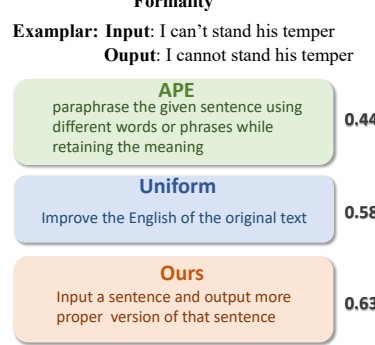

Figure 8: **Comparison** of the best instructions in Formality task, which aims to rephrase the sentence in formal language.

APE selects "Sort the inputs alphabetically and then output the first, third, fifth, and seventh elements of the sorted list." as its top instruction, which is not precise at all. In contrast, INSTRUCTZERO optimized instruction "Find a list of the animals from the input list" is clearer. Another example of the "Formality" task in Fig. 8 also demonstrates that INSTRUCTZERO can better comprehend the exemplars and yield more precise instructions.

## 5 RELATED WORK

**Large Language Models.** The scaling up of transformer-based language models (Vaswani et al., 2017; Devlin et al., 2018) has consistently improved performance across various downstream NLP tasks. As a consequence, numerous capabilities of large language models (LLMs) have been uncovered, encompassing few-shot in-context learning (Brown et al., 2020), zero-shot/few-shot sequential reasoning (Kojima et al., 2022; Wei et al., 2022), and the automatic generation of intructions (Honovich et al., 2022). In this paper, we study how to guide open-source LLMs to generate and improve instructions for subsequent API LLMs. Experiments demonstrate that INSTRUCTZERO has the potential to break the scaling law of LLMs: a $10\times$ smaller open-source model (Vicuna) can be used to optimize an instruction with superior performance compared to a much larger LLM (ChatGPT used in APE).

**Instruction-following and instruction-finetuning.** LLMs are able to follow instructions, a capability that can be reinforced by instruction tuning (Chung et al., 2022; Iyer et al., 2022; Sanh et al., 2021), e.g., finetuning the model on a wide range of tasks using human-annotated prompts and feedbacks (Ouyang et al., 2022), or supervised finetuning using public benchmarks and datasets (Wang et al., 2022). ChatGPT is well-known as an instruction follower but is a black-box model. Vicuna [5] finetunes the open-source LLaMA (Touvron et al., 2023) using only 700K instruction-following examples from user-shared ChatGPT data (OpenAI, 2023), which exhibits similar instruction-following capability as ChatGPT. Zero-shot learning does not allow finetuning the LLM or training an adapter (Hu et al., 2021). Moreover, for black-box LLMs, any model training is infeasible. In these cases, we can only improve the downstream task performance by optimizing the instruction, which is exactly the problem addressed by INSTRUCTZERO and is a challenge complementary to instruction finetuning.

**Prompting and Auto-Prompt.** Prompting prepends some soft token embeddings, textual instruction, or/and input-output exemplars of a target task to the original input query as context information to

---

[5]https://vicuna.lmsys.org/

guide the reasoning of LLMs. Soft prompts as differentiable are learnable and can be optimized by backpropagation (Li & Liang, 2021; Lester et al., 2021; Liu et al., 2021; Chen et al., 2023c;b). However, API LLMs are black boxes that only allow hard prompts in natural languages, whose optimization is challenging due to the combinatorial and highly structured search space. (Deng et al., 2022) relies on reinforcement learning (RL) to optimize hard prompts while INSTRUCTZERO optimizes an instruction in the output space of an open-source model $g(\cdot)$ without RL by applying BO of a soft prompt to $g(\cdot)$. Another line of works of prompting (Brown et al., 2020) relies on the generative power of LLMs and asks them for self-debugging (Chen et al., 2023d) or self-improve (Huang et al., 2022). Auto-prompt (Shin et al., 2020) conducts a gradient-guided search in a pre-defined set of triggers to build up prompt automatically. APE (Zhou et al., 2022) adopts a black-box LLM such as GPT-3 to generate instructions and select better ones but its search in the instruction space can be inefficient without exploiting the correlation between the evaluated instructions, which may lead to sub-optimal results. Compared to them, INSTRUCTZERO leverages open-source models to generate instructions to explore and thus does not need a predefined set of triggers.

**Bayesian Optimization.** Over the last decade, Bayesian optimization (BO) (Frazier, 2018) has emerged as a highly effective black-box optimization approach in various domains such as drug and molecule design (Gómez-Bombarelli et al., 2018; Jin et al., 2018; Kajino, 2019). Since our goal is to optimize instructions for a black-box LLM, it is akin to the BO in combinatorial spaces (Gómez-Bombarelli et al., 2018), which is challenging especially when the space is highly structured. Recent approaches (Kajino, 2019; Jin et al., 2018; Lu et al., 2018) study to reduce the combinatorial black-box optimization to BO in a latent space, given a mapping from the latent space to the combinatorial space learned by deep generative models (DGMs). LADDER (Deshwal & Doppa, 2021b) introduces structure-coupled kernels to align the abundant information of each structure in the combinatorial space with its corresponding representation in the latent space. In a similar vein, our instruction-coupled kernel aims to align the soft prompt kernel with the similarity between instructions. However, our kernel has a different form and aims to guide the open-source LLM to explore different soft prompts and generate better instructions.

## 6 DISCUSSION, CONCLUSIONS, AND LIMITATIONS

In this paper, we propose INSTRUCTZERO, an efficient zeroth-order instruction optimization method that can improve the zero-shot learning and instruction-following of black-box LLMs with only API access. INSTRUCTZERO addresses the crucial challenge of prompt engineering, which is a combinatorial black-box optimization that currently still relies on human expertise and costly experience. In contrast, INSTRUCTZERO can automatically optimize and generate human-readable and task-relevant instructions for arbitrary tasks by leveraging the in-context learning and generative power of recent open-source LLMs. Its key idea is to optimize a soft prompt that guides an open-source LLM to generate instructions for the black-box LLM to address the task. The zero-shot performance on the task using different soft prompts is collected by a Bayesian optimizer to improve the soft prompt progressively. In this way, INSTRUCTZERO overcomes the combinatorial challenge and reduces the original instruction optimization to an efficient latent space BO.

We provided visualizations of the optimization trajectories, optimized instructions, an ablation study, and extensive comparison to other auto-instruction approaches on 32 tasks. INSTRUCTZERO using a small Vicuna model outperforms non-optimization methods that utilize a much larger and more powerful LLM for instruction generation. As a general instruction optimization tool, INSTRUCTZERO can be used to improve the efficiency of human-AI interactions through APIs of black-box models and enhance the downstream task performance of these models without any model finetuning.

However, the application of INSTRUCTZERO in current experiments does not include more complicated tasks requiring refinement, multi-step planning, or human interactions, e.g., cooking recipe, website design, trip planning, and booking, etc. Improving the efficiency of solving these tasks by instruction optimization can potentially save more costs.

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

## A  SUPPLEMENTARY MATERIAL

In Table 2, we report the best instruction generated by INSTRUCTZERO for each task and the associated performance (execution accuracy). In Table 3, we report the task description and demos for the 8 new tasks used in our paper. (the other 24 tasks are the same as the ones used in APE (Zhou et al., 2022)).

## B  FREQUENTLY ASKED QUESTIONS

### B.1  WHY IS THE PERFORMANCE OF APE QUITE POOR ON CHATGPT?

In the practical setting, we only have access to the textual output from the black-box LLM, e.g., ChatGPT. So we could not calculate the log probability as the score function in INSTRUCTZERO (ours) as original APE (Zhou et al., 2022). We provide our code for reproducing the experimental results using ChatGPT as black-box LLM.

### B.2  CODE AVAILABILITY

We include our code in the file "INSTRUCTZERO" so reviewers are able to reproduce our results.

### B.3  CHOICES OF KERNEL IN BAYESIAN OPTIMIZATION

We investigate how the Instruction-Coupled Kernel affects the final performance of INSTRUCTZERO. We ablate the effective of Instruction-Coupled Kernel by removing the instruction component, namely Standard Kernel. Specially, we only consider the structure of latent space, kernel 9 can be rewritten:

$$\boldsymbol{K}_{i,j} = k(\boldsymbol{p}_i, \boldsymbol{p}_j) = \boldsymbol{l}_i^\top \boldsymbol{L} \boldsymbol{l}_j. \tag{10}$$

Table 4 shows the Instruction-Coupled Kernel outperforms the Standard Kernel, indicating the effectiveness of Instruction-Coupled Kernel in our method.

### B.4  OPTIMIZATION PROCESS ON MORE TASKS

Fig. 9, as a supplementary of Fig. 5, presents how the zero-shot accuracy (for the top 15% of instructions facilitated by our algorithm) is improved over the instruction optimization iterations of INSTRUCTZERO. For the majority of evaluated tasks, INSTRUCTZERO achieves a consistent uptick in accuracy, indicating an effective and efficient optimization process by our black-box instruction optimization approach.

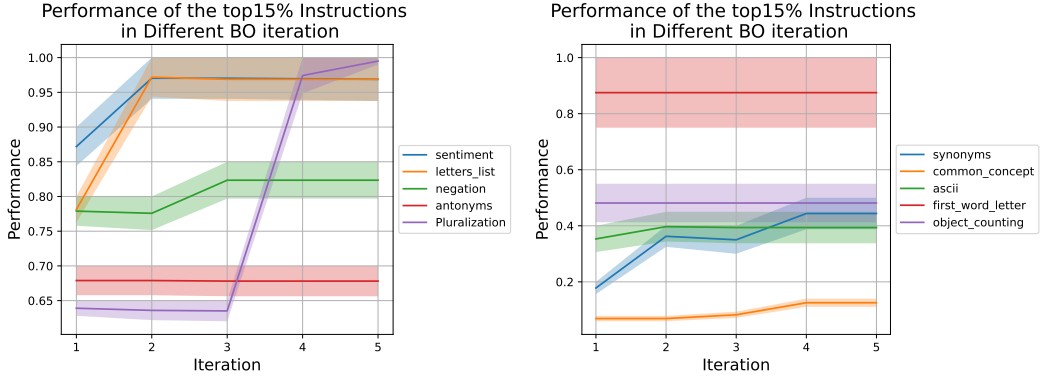

Figure 9: Supplementary results: Top-15% instructions after every iteration (1-5) of INSTRUCTZERO on different tasks.

## C  EVALUATION METRICS

**Exact Match (EM):** When evaluating each question and answer pair, if the model's predicted response precisely aligns with any of the correct responses, $EM = 1$. If it doesn't align perfectly, $EM = 0$.

Tasks using metric "EM": Passivation, Antonyms, Diff, First letter, Letters List, Negation, Num2Verbal, Rhymes, Second Letter, Similarity, Sentiment, Pluralization, Sum, Translation-En_De, Translation-En_Es, Translation-En_Fr, Second Word.

**Exact Set (ES):** When evaluating each question and answer pair, if the model's predicted response precisely aligns with the correct responses set, $ES = 1$. If it doesn't align perfectly, $ES = 0$.

Tasks using metric "ES": Orthography, Taxonomy.

**Contain:** If the characters in the model's predicted answer are part of the characters in the correct responses, $Contain = 1$. If it doesn't align perfectly, $Contain = 0$.

Tasks using metric "Contain": Ascii, Debugging, CS Algorithm, Object Counting, Synonyms, Unscrambling, Word Sorting.

**F1:** The F1 score is calculated by comparing individual words in the predicted response to those in the actual or True Answer. The common words between the predicted and actual answers form the basis for the F1 score. Precision is determined by the proportion of common words to the total words in the predicted response, while recall is calculated as the proportion of common words to the total words in the actual answer.

Tasks using metric "F1": Common, Formality.

## D  DIFFERENT COMBINATIONS OF API LLM + OPEN-SOURCE LLM

We have conducted further experiments exploring a variety of combinations between API-based LLMs and open-source LLMs. Specifically, in addition to our experiments with Vicuna+ChatGPT combinations, we also include GPT-4 and WizardLM (Xu et al., 2023) as the open-source LLM and API LLM, respectively. The results on "Second Letter" and "Cause Selection" tasks are reported in Tab. 5 and Tab. 6, which show the effectiveness of our algorithms on different combinations of API LLM and open-source LLM. In these two tables, we also include the human instructions, which are obtained from (Honovich et al., 2022). Notably, the instructions generated by our algorithms could be significantly better than the human instructions.

## E  COMPARISON OF INSTRUCTZERO INSTRUCTIONS AND HUMAN INSTRUCTIONS

We show the comparison of InstructZero instructions and human instructions in Tab.7. The comparison shows that InstructZero can produce much better instructions than human instructions.

| Dataset | Best Instruction | Performance |
|---|---|---|
| Unscrambling | Find words that are anagrams of each other | 0.67 |
| Letters List | Input 'matter' and get 'm a t t e r' as output | 1.0 |
| Debugging | Input the code and the output would be shown | 0.50 |
| Word Sorting | make a code that takes an input of a list and produces an output that is the list with each word in the list in alphabetical order. | 0.64 |
| Cause Selection | Give a positive or negative output depending on the input | 0.86 |
| Antonyms | Make the pairs of words opposite. | 0.89 |
| Categorization | Create a system which could understand what the inputs and outputs were, and then use that knowledge to fill in the blanks in the following sentence: Input: Togo, Eritrea, and Burundi Output: African countries. The system would then use this knowledge to fill. | 0.35 |
| Larger Animal | Remove the input that has the smaller animal and keep the larger animal | 0.91 |
| Sum | Find the sum of the two input numbers | 1.0 |
| Periodic | Create a new element using the periodic table. | 1.0 |
| Passivation | Make the sentences more natural by flipping the subject and verb | 1.0 |
| Common | Make the output related to the input in some way | 0.15 |
| Odd one out | Determine the word that is different. | 0.92 |
| Diff | Find the difference between the two numbers | 1.0 |
| Ascii | Make the letters appear in the correct order. | 0.33 |
| Object Counting | create a program that takes an input (a list of things) and outputs the number of things in the list | 0.48 |
| Negation | Swap the truth value of the input statements with the opposite of the truth value | 0.80 |
| First Letter | Find the first letter of each word in the list | 1.0 |
| Second Letter | Create a function that takes a string as input and returns the first character that is a vowel. | 0.62 |
| Formality | Input a sentence and the output would be a more proper version of that sentence. | 0.63 |
| CS algorithm | Generate a string which is the input to the function above, which when processed will give the output below. | 0.38 |
| Negation | Swap the truth value of the input statements with the opposite of the truth value | 0.80 |
| Pluralization | Make plural words from the input words | 1.0 |
| Rhymes | Write a function that takes a word as input and returns the output word | 0.46 |
| Num2Verbal | Write a function that takes an integer as input and returns the number in words | 1.0 |
| Similarity | Find the difference between the two sentences and the output was 4 - almost perfectly | 0.19 |
| Taxonomy | Create a program that generates a list of animals based on the input provided | 0.82 |
| Sentiment | Generate a short review based on the sentiment of the user but the output was always positive or negative | 0.93 |
| Orthography | Input a sentence and the output would be a word from the sentence | 0.51 |
| Synonyms | Create a list of words that have a similar meaning | 0.38 |
| Translation EN-DE | Translate the English words to German | 0.84 |
| Translation EN-ES | Take the input text and translate it into Spanish. | 0.87 |
| Translation EN-FR | Convert all of the words in the input column to their French translations. | 0.89 |

Table 2: The best instruction found by INSTRUCTZERO.

| Name | Demos | Description |
|---|---|---|
| CS Algorithm | Input: XDWO XDWOHDGYT
Output: 4 | Given two strings, determine the length of the longest substrings |
| Unscrambling | Input: ilpf
Output: flip | common sense, gender bias, many-shot multiple choice |
| Categorization | Input: Shaymin, Chatot, and Reshiram
Output: Pokeman | Categorize the input list. |
| Periodic | Input: 42
Output: molybdenum | Write the periodic element based on the input number. |
| Odd one out | Input:Monday, spring, summer, winter
Output:Monday | common sense, gender bias, many-shot multiple choice |
| Ascii | Input: .._..._..._..._..._.. ./../../../../..
(.b.l.r.l.o.l.k.l.e.) ._/._/._/._/._/
Output: broke | What word is displayed in the ASCII art below? |
| Object Counting | Input: I have a duck, a mouse, three pigs, two fish, and a donkey.
Output: 8 | Count the objects in the input. multiple choice |
| Debugging | Input: print('1' + 2)
Output: TypeError: must be str, not int | Debug the input program. |

Table 3: The description, demos of the 8 new tasks. The other 24 tasks are the same as APE (Zhou et al., 2022).

| Task | Instruction-Coupled Kernel | Standard Kernel |
|---|---|---|
| Sentiment | 0.93 | 0.83 |
| Negation | 0.80 | 0.39 |
| Larger Animal | 0.91 | 0.81 |
| Second Letter | 0.62 | 0.33 |
| Formality | 0.63 | 0.44 |
| Debugging | 0.50 | 0.25 |
| Unscrambling | 0.58 | 0.67 |
| Odd one out | 0.92 | 0.9 |
| Ascii | 0.33 | 0.13 |
| CS algorithm | 0.38 | 0.26 |

Table 4: **Ablation study.** Performance (higher is better) of different kernels (1) Instruction-Coupled Kernel proposed in our paper (2) Standard Kernel only using the structure of latent space.

| Task: Second Letter | Best Instruction | Acc |
|---|---|---|
| Human instruction + ChatGPT | write the second letter of the input | 0.88 |
| Human instruction + GPT-4 | write the second letter of the input | 0.96 |
| Vicuna-13B + ChatGPT | Create a function that takes a string as input and returns the first character that is a vowel. | 0.62 |
| Vicuna-13B + GPT-4 | Take a string as an input and returns the second letter of the input string. | 0.99 |
| WizardLM-13B + ChatGPT | Create a function that takes a string as an input and returns the second letter of the input string. | 0.99 |
| WizardLM-13B + GPT4 | Remove the first letter of the input words and output the second letter. | 1.0 |

Table 5: More evaluation results on second letter tasks. We not only use ChatGPT, GPT-4 as our API LLMs but also include WizardLM (Xu et al., 2023), Vicuna as our "second letter" task requires models to output the second letter of the input word, e.g., for input "multilingual", the output should be "u".

| Task: Cause Selection | Best Instruction | Acc. |
|---|---|---|
| Human instruction + ChatGPT | decide which event occurred first | 0.52 |
| Human instruction + GPT-4 | decide which event occurred first. | 0.72 |
| Vicuna-13B + ChatGPT | Give a positive or negative output depending on the input | 0.86 |
| Vicuna-13B + GPT-4 | Determine the relationship between the two sentences and identify which sentence is the main cause | 1.0 |
| WizardLM-13B + ChatGPT | create a function that takes two sentences as input and returns the second sentence if the first sentence is not the cause of the second sentence. If the first sentence is the cause of the second sentence, the function should return an empty string. | 0.58 |
| WizardLM-13B + GPT-4 | Identify the cause and effect relationship between two sentences and provide the cause sentence as the output | 0.76 |

Table 6: More evaluation results on the Cause Selection task.

| Task | Human Instruction | Score | Score(Ours) |
|---|---|---|---|
| Active to passive | Write the sentence from the other point of view | 0.69 | 1.0 |
| Cause Selection | decide which event occurred first | 0.52 | 0.86 |
| Taxonomy | Write all the animals in the input in a random order | 0 | 0.82 |
| Translation EN-DE | Translate the word to German | 0.74 | 0.84 |

Table 7: Comparison of InstructZero instructions and human instructions. For the instructions obtained by our algorithm, please refer to Tab. 2.

