# OpenReview forum: "InstructZero: Efficient Instruction Optimization for Black-Box Large Language Models"
_ICLR.cc/2024/Conference — Submitted to ICLR 2024_

### Official Review · Reviewer_FwK3 · 2023-10-31

**Soundness:** 2 fair
**Presentation:** 4 excellent
**Contribution:** 2 fair
**Rating:** 6
**Confidence:** 3

**Summary:**

The paper presents an innovative approach to identifying the "optimal" instruction for large language models with the aim of improving generative quality. This work falls under the burgeoning area of prompt search methods, which have gained significant attention recently, exemplified by methods such as APE, RLPrompt etc. Unlike conventional methods that optimize discrete instructions, the authors propose optimizing a low-dimensional "soft prompt" using dimensionality reduction. The optimized soft prompt is applied to an open-source Lifelong Learning Model (LLM) to generate instructions for a black-box LLM. The optimization process is iterative, involving zero-shot evaluation of the black-box LLM's performance, which is then used in a Bayesian optimization scheme to refine the soft prompts. This iterative process continues until convergence. Experimental results on the BBH benchmark show that the proposed method yields superior performance across all 32 tasks.

**Strengths:**

- The proposed methodology is both innovative and well-explained, making a valuable contribution to the area of prompt optimization in large language models.
- The empirical results are compelling, demonstrating superior performance across all 32 tasks on the BBH benchmark.
- The use of Uniform as a comparative baseline effectively underscores the benefits of the proposed iterative Bayesian Optimization (BO) process.

**Weaknesses:**

- The paper could benefit from a broader evaluation scope. Considering additional tasks such as reasoning QA GSM8K, machine translation, or summarization could provide a more comprehensive view of the method's effectiveness.
- The inclusion of only two comparative baselines, APE and Uniform, limits the robustness of the evaluation. Expanding the set of comparative baselines could provide a more holistic understanding of the method's performance relative to existing work. For example, RLprompt and Autoprompt are also two good prompt search methods.
- The paper presents a puzzling result related to APE's performance, which is reported to have only a 0.04 accuracy in Figure 1. Upon closer inspection, it becomes apparent that the original APE experiments were based on instructgpt, where the prediction probability could be obtained. While this paper employs the more powerful Turbo 3.5 API, which cannot access the prediction  probability. Thus I think the comparison here is not fair enough, as APE is a much weaker version than the original paper. This discrepancy introduces confusion and could affect the perceived validity of the comparative results.  This drawback again highlights a more fair comparison is required, e.g., other prompt search baselines are needed. Also, what about the comparison with zero-shot COT, 'Please Think step by step'?
- There is ambiguity in the claim about the method being applicable for zero-shot evaluation. While it's true that the proposed method employs a black-box LLM API for zero-shot generation, the Bayesian Optimization (BO) process requires labeled data. This seems to contradict the zero-shot claim and may constitute an overstatement.

**Questions:**

- I wonder why the results of APE are so weak in Figure 1.

---

> ### Author Response · Authors · 2023-11-17
>
> **Q1:** The paper could benefit from a broader evaluation scope, e.g., GSM8k, machine translation
>
> **A1:** Thanks for your suggestion! We add new experiments of GSM8K, AQUA, and SVAMP by evaluating the zero-shot performance. Following [1], the reasoning template is designed as `I have some instruction examples for solving school math problems. Instruction: Let’s figure it out! Instruction: Let’s solve the problem. Instruction: Let’s think step by step. Write your new instruction that is different from the examples to solve the school math problems. Instruction:` The results are reported in the table below:
> |Dataset| Method | Instruction | Results |
> |---| ---    | ---        | ---|
> |GSM8k| CoT    | Let's think step by step | 0.718 |
> |   | Ours   | Let's use the instruction to solve the problem | 0.743|
> |AQUA| CoT    | Let's think step by step | 0.511 |
> |   | Ours   | Let's break down the problem | 0.543|
> |SVAMP| CoT    | Let's think step by step | 0.763 |
> |   | Ours   | Let's use the brain | 0.795|
>
> [1] USE YOUR INSTINCT: INSTRUCTION OPTIMIZATION USING NEURAL BANDITS COUPLED WITH TRANSFORMERS. https://arxiv.org/abs/2310.02905

---

> ### Author Response · Authors · 2023-11-17
>
> **Q2:** Comparison with other prompt search methods, e.g., RLprompt.
>
> **A2:** We discussed the key difference with the RLprompt in our related work section: [2] (RLprompt) relies on RL to optimize hard prompts while InstructZero optimizes an instruction in the output space of an open-source model $g(\cdot)$ **without RL** by BO of a soft prompt to g(·). Moreover, the instructions found by our method are all fluent while the instructions found by RLprompt are not, e.g., in Yelp Review (a text classification task), RLprompt found `StaffAreaFocusHardware Advisory` as their best prompt. Hence, InstructZero is a better pipeline to optimize human-readable instructions, which is more challenging than the task of RLprompt.
>
> **Q3:** There is ambiguity in the claim about the method being applicable for zero-shot evaluation.
>
> **A3:** Thank you for your feedback! We did include a few labeled data as input demos for the open-source models during instruction optimization. However, the inference and evaluation of the optimized instructions on the black-box LLMs are zero-shot. Hence, the application of the optimized instructions is indeed zero-shot and it does not overclaim. We will make this clear in the polished manuscript.
>
>
>
>
> [2] RLprompt: Optimizing Discrete Text Prompts with Reinforcement Learning. https://arxiv.org/pdf/2205.12548.pdf

---

> ### Author Response · Authors · 2023-11-21
>
> Dear Reviewer,
>
> We haven't heard from you since sending you our rebuttal. Since we are approaching the last day of the reviewer-author discussion, it would be really nice of you to confirm if your initial concerns (most of them are clarification questions) were successfully addressed by our rebuttal. We are looking forward to your feedback and we kindly expect that you can raise the score if all your main concerns have been resolved.
>
> Thanks!
>
> Best regards,
>
> Authors

---

> > ### Comment · Reviewer_FwK3 · 2023-11-22
> > **After reading the rebuttal**
> >
> > I would like to extend my sincere appreciation to the authors for their efforts in addressing the concerns raised during the rebuttal process. I am pleased to note that most of my concerns have been adequately addressed. However, I still have reservations regarding the weak score of APE, and I also observed that there was no response to the W3 concern. It's possible that I may have overlooked pertinent information or misunderstood the issue at hand.
> >
> > After careful consideration, I have decided to marginally increase my score to 6.

---

> > > ### Author Response · Authors · 2023-11-22
> > > **Thank you! Response to W3 (with experiments) is in General Response**
> > >
> > > Thank you for letting us know that most of your concerns have been addressed! We appreciate you for raising your score! Your constructive input remains invaluable to us, and we appreciate your dedication to enhancing the quality of our manuscript. Thank you for your time and consideration.
> > >
> > > Every concern of you matters a lot to us and we carefully addressed every one of them. **Our response to W3 regarding the weak score of APE can be found at the very beginning of our general response (at the very top of this page)**. This is due to the number of demos (i.e., 5 demos, compared to 100 in the original APE paper) used in both APE and our method (for a fair comparison in a challenging setting). We also provided an experimental comparison between APE and our method when using more demos (100 demos for APE, 10 demos for ours). Our method still finds better instructions than APE even using only 10% of demos used by APE.
> > >
> > > Would you mind confirming if this addressed your concern in W3? If you have any further questions, please let us know and we are happy to discuss and elaborate on more details. Thank you for reading our responses and providing valuable feedback! If this reply has addressed your concerns in W3, we would really appreciate it if you would consider raising your score.

---

### Official Review · Reviewer_4arY · 2023-11-01

**Soundness:** 2 fair
**Presentation:** 3 good
**Contribution:** 2 fair
**Rating:** 3
**Confidence:** 4

**Summary:**

This paper proposes a method to optimize the instructions for black-box large language models. The proposed method uses an open-source LLM to convert a soft prompt to an instruction, and then uses the instruction as input to the black-box LLM. Bayesian optimization is then used to optimize the soft prompt, which can iteratively propose new soft prompts and instructions to be evaluated by the black-box LLM.

**Strengths:**

- The proposed method of using another open-source LLM to help convert a soft prompt to an instruction and then using the instruction as input to the black-box LLM is an interesting and intuitive idea.
- The graphical illustrations in Figure 2 and 3 are nice and helpful for understanding.
- The results in Figure 4 indeed show that the proposed method improve over APE and Uniform.

**Weaknesses:**

- One overall observation from the experimental results which concerns me is that it seems that APE does not consistently perform better than Uniform? Both Figure 4 and Figure 1 seem to suggest this, for example, in Figure 1, the improvement over APE seems to be larger than over Uniform. This is an unexpected observation and I think should be explained, because it may suggest that performances of APE might be underestimated in the experiments here.
- I have some questions and concerns about the instruction-coupled kernel. First of all, it seems that to calculate this kernel between a pair of input soft prompts, you need to have the evaluated scores for both soft prompts (correct me if I'm wrong)? If this is the case, then when you calculate the vector $\boldsymbol{k}$ in equations 4 and 5, this instruction-coupled kernel cannot be used to calculate these kernel values and therefore these kernel values will simply use the normal squared exponential or matern kernel? In this case, I wonder how much this instruction-coupled kernel actually helps the performance of the Bayesian optimization, because the vector $\boldsymbol{k}$, which directly measures the distance between a new soft prompt and other previously evaluated soft prompts and therefore has a huge influence on the uncertainty measure, cannot make use of it. I see that you have an ablation study in Table 4 to show the effect of using the instruction-coupled kernel, but why did you only show the comparison for a small number of selected tasks? I think to see whether this kernel is actually useful, it's important to fairly run this ablation study in all tasks and make an overall comparison.
- About the ablation study (Section 4.3), it looks like the scores "w/o Manual" is in general better than "Manual"? This is also puzzling because it implies that the meta-prompt used by APE may not be useful...
- The proposed method InstructZero seems to only optimize the zero-shot performance of the instructions instead of few-shot performance. However, since you already have access to these input-output exemplars which are used as input to the open-source LLM, why don't you also use them as input to the black-box LLM to improve the performance? So this may bring into question how practical the experiments are.
- (minor) Equations 4 and 5, it seems that the matrix $K$ is not explained.

**Questions:**

My questions are listed under "weaknesses" above.

---

> ### Author Response · Authors · 2023-11-17
>
> **Q1** questions and concerns about the instruction-coupled kernel.
>
> **A1:** $\mathbf K$ in Eq. (4)-(5) are computed using Eq. (9). It depends on both the soft prompts' similarity $l(\cdot,\cdot)$ and their generated instructions' similarity $s(\cdot,\cdot)$. Since its computation depends on the similarity between explored soft prompts $\{p_1, ..., p_m\}$ from previous iterations and their evaluation scores $\{H(p_1), ..., H(p_m)\}$, $K$ can be predetermined before computing Eq. (4)-(5). $k(p, p_i)$ and $k(p,p)$ of Eq. (4)-(5) only measure the kernel similarity in the soft prompt space using $l(\cdot,\cdot)$ so we do not need to evaluate every $p$'s performance during optimization.
>
>
> Instruction-coupled kernel $\mathbf K$ aims to align the exploration in the soft prompt space (what the BO directly does in InstructZero) and the optimization of the textual instruction (which is the final goal). To this end, the kernel in BO is expected to reflect the similarity of the generated instructions for the target task. So we apply the instruction-coupled kernel in computing $\mathbf K$. However, to compute $\mathbf k$, we cannot afford to evaluate every soft prompt $p$'s generated instruction during the maximization of the acquisition function and its evaluation is not differentiable. So in practice we only apply the soft prompt space kernel $l(\cdot,\cdot)$ in computing $k(p, p_i)$ and $k(p,p)$ of Eq. (4)-(5).
>
> In all our experiments, the above approach performs promisingly and we haven't observed any degradation of using the kernel.

---

> ### Author Response · Authors · 2023-11-17
>
> **Q2** "w/o Manual" is in general better than "Manual"? This is also puzzling because it implies that the meta-prompt used by APE may not be useful.
> **A2:** We agree that the meta-prompt used by APE may not be useful. The effectiveness of the meta-prompt may vary on models or tasks. For example, an effective meta-prompt for GPT-3, as in the experiments of the APE paper, may not be the best on other models.
>
> **Q3:** Why don't use the demos which have already been used for finding the instructions as the input to the black-box LLM to improve the performance?
> **A3:** For a fair comparison, we follow the APE pipeline, which only uses these input-output examplars for instruction inductions and conducts zero-shot evaluations of the induced instructions.

---

> ### Author Response · Authors · 2023-11-21
>
> Dear Reviewer,
>
> We haven't heard from you since sending you our rebuttal. Since we are approaching the last day of the reviewer-author discussion, it would be really nice of you to confirm if your initial concerns (most of them are clarification questions) were successfully addressed by our rebuttal. We are looking forward to your feedback and we kindly expect that you can raise the score if all your main concerns have been resolved.
>
> Thanks!
>
> Best regards,
>
> Authors

---

> > ### Comment · Area_Chair_xDTG · 2023-11-22
> >
> > Dear Reviewer 4arY,
> >
> > Please do consider responding to the authors soon regarding your remaining concerns.
> >
> > Thanks!

---

### Official Review · Reviewer_6Tou · 2023-11-02

**Soundness:** 3 good
**Presentation:** 4 excellent
**Contribution:** 3 good
**Rating:** 8
**Confidence:** 3

**Summary:**

This paper proposes to use Bayesian optimization to learn an instruction with an open-source LLM so that the instruction improves the zero-shot results of a black-box LLM. Since instructions are discrete, this work instead iteratively learns a small soft prompt which then gets decoded as an instruction. Each updated instruction is evaluated on the black-box LLM whose training accuracy is used to find a better soft prompt.

**Strengths:**

This is an interesting direction toward automating prompt engineering for API models, and shows strong results.

**Weaknesses:**

It would be equally interesting to see qualitative analysis of the errors and various failures modes by the method and the different components used for optimization (e.g. open-source/black-box LLMs).

**Questions:**

Some of the similarity metrics are chosen because black-box models don't necessarily return the log-probs. An ablation could have been run where an open-source model is used for both instruction proposal and loss evaluation. Then, we have access to log-probs and/or gradient and will have a better understanding of how much performance we are losing. Could be interesting, not saying this should have been run.

---

> ### Author Response · Authors · 2023-11-17
>
> **Q1:** Some of the similarity metrics are chosen because black-box models don't necessarily return the log-probs. An interesting study could be using an open-source model for both instruction proposal and loss evaluation
>
> **A1:** Thanks for your suggestions! We will continue to polish our work as you suggested.

---

### Author Response · Authors · 2023-11-17
**General Response**

We appreciate all reviewers for providing useful feedback! We hope our responses address your concerns. Please feel free to raise further concerns and we are open to discuss.

Reviewers #FwK3 and #4arY both point out that the performances of APE might be underestimated in our experiments. We would like to clarify that the original APE can access 100 demonstrations~(input-output pairs) for instruction optimization. In every iteration of APE, they randomly draw 5 new demos from a hold-out set of 100 demos. Instead, our hold-out set contains only 5 demos and InstructZero does not have access to more new demos---this is more practical and also more challenging. In our experiments, for a fair comparison, both APE and InstructZero have access to the same hold-out set for instruction induction and use GPT-3.5-turbo as the API LLMs.

In addition to the reported 5-demo experiments, we expand InstructZero's hold-out set of demos to 10 demos and compare it with the original APE using a hold-out set of 100-demos. We keep the API LLM the same as the one used in our original manuscript, i.e., ChatGPT. The results are shown as follows:
| Dataset | APE (5 demos) |APE(100 demos) | Ours (5 demos) |    Ours(10 demos) |
| -----   |  ---          | ---           | ---            |   ----            |
| Negation|  0.76         |   0.83       |   0.80           |  0.86  |
| Synonym |  0.37         |       0.37      |   0.42            |    0.46 |
| Second Letter |  0.58  | 0.95 | 0.63 | 1.0 |

As shown on three tasks, InstructZero finds better instructions than APE even using only 10\% demos used by APE.

---

### Meta-Review · Area_Chair_xDTG · 2023-12-05

**Metareview:**

This paper proposes to optimize the instruction in the prompt using Bayesian optimization.


**STRENGTHS**

(1) The reviewers agree that the proposed method is interesting and new.

(2) The proposed method outperforms the tested baselines.


**WEAKNESSES**

The authors have adequately resolved the concern with underestimating the performances of APE in their rebuttal.

After reviewing the authors' rebuttal, a number of major concerns remain, as discussed below. The review provided by Reviewer 4arY (who has responded in the AC-reviewer discussion phase) has provided strong evidence of these concerns, while the review given by reviewer 6Tou seems to have lacked detailed arguments in favor. Recommendations are provided for some of them to improve this paper.

(1) The authors' rebuttal did not adequately address the concern on $\mathbf{k}$ in Eq. 4-5. Computing $\mathbf{k}$ directly using eq. 9 does not seem possible in practice. However, if we take the authors' response at face value, does it mean that they only use $l$ to compute $\mathbf{k}$ but not $\mathbf{S}$ in equation 9? This point seems to be re-iterated in the last sentence of the 2nd paragraph in A1. How then would the kernel remain valid when $\mathbf{K}$ is defined using eq. 9 but $\mathbf{k}$ is defined differently? This technical ambiguity needs to be addressed by the authors. Hence, the ablation study needs to be conducted on all tasks instead of a small number of them.

(2) Though the authors have rebutted with the reason of fairness in comparison with APE, they can use the input-output exemplars to assess how much the performance of InstructZero can be improved, since they are already assumed to be available in the problem setting. Furthermore, as mentioned by a reviewer, APE has also performed few-shot demonstrations.

(3) The authors have used the same reasoning template as that in [1]. Hence, we assume that the authors are aware of similar results reported in [1] (Table 3). The authors have not discussed why the results are similar or different and whether they have taken some of the results from [1] or run the corresponding experiments. For example, with the score of 0.795 for SVAMP, the authors obtain the instruction "Let's use the brain", but it shows "Let's use the equation" in [1] for the authors' proposed method.

(4) Though the authors have described the qualitative differences with RLPrompt in the related work section, it is not obvious that RLPrompt would necessarily perform worse than their proposed method empirically.

(5) The authors say that "In experiments, we apply a mini-batch version of INSTRUCTZERO that explores 25 soft prompts in every iteration. The only major change required is to select the top-25 soft prompts with the largest u(p)". This does not seem like an effective way of finding a good batch of soft prompts since they can be clustered near to each other in the soft prompt space due to their large u(p). The authors are strongly encouraged to look into the literature of batch Bayesian optimization algorithms to understand how to construct an effective batch.

The authors are strongly encouraged to address the above concerns and consider the reviewers' feedback when revising their paper.

**Justification For Why Not Higher Score:**

Major concerns for this work remain, as listed in the weaknesses above. The cons outweigh the pros.

**Justification For Why Not Lower Score:**

N/A.

---

### Decision · Program_Chairs · 2024-01-16

Reject